# [Re] Interactive Two-Stream Decoder for Accurate and Fast Saliency Detection

1    ## Reproducibility Summary

2    *This article describes our attempts to reproduce the work presented in the paper - Interactive Two-Stream Decoder for*
3    *Accurate and Fast Saliency Detection which published in CVPR2020, as part of the ML Reproducibility Challenge*
4    *2020. Our work consists of three parts: (1) evaluation using the provided pre-trained models from the original paper;*
5    *(2) evaluation using our retrained models; (3) parameters analysis.*

6    **Scope of Reproducibility**

7    There are two main objectives of our report: (1) evaluation using the provided *pre-trained* models from the original
8    paper; (2) evaluation using our *retrained* models

9    **Methodology**

10   We used publicly available source code provided by the authors. Minor changes were made to the source code in order
11   to load the model weight properly. The reproducibility experiments followed the protocal as described in the original
12   paper. We performed the training on a machine with two GTX2080TI GPUs. The training and validation took about 5
13   and 7.5 hours for ResNet-based and VGG-based models, respectively.

14   **Results**

15   The reproducibily using authors' pretrained model was of no success and we recorded the discrepancy of as high as
16   11.4%. While for the retrained models, the difference was observed to be as high as 8.6%. Therefore, we carried out
17   statistical test to further confirm the reproducibility of the investigated work. We failed to reject the null hypothesis,
18   which implies that there is no signfiicant difference between the reported results and our results.

19   **What was easy**

20   The pretrained models have been shared by authors. One can try out the models with little effort as minor changes to
21   the name of the parameter keys of the provided PyTorch models are expected.

22   **What was difficult**

23   We observed some differences between the reported results. We spent longer time to look into the issue and also had
24   discussion with the authors. We repeated the experiments several times for each model to ensure the correctness of
25   our results. Despite all the challenges, we finally come to the conclusion that the investigated work is reproducible, as
26   evidenced by the statistical test.

27   **Communication with original authors**

28   We raised and discussed some issues via GitHub. We also discussed some technical details and also about the result
29   discrepancy with the authors via emails.

# 1 Introduction

Saliency detection is a mechanism to detecting visually attentive objects or regions in an image. To better discover the visually distinctive objects or regions, the saliency detection methods are expected to understand both the global and local features Qin et al. [2019]. The global features are essential to provide high-level semantic information for identifying salient objects while the local features are exploited for features refinement. The investigated paper – Interactive Two-Stream Decoder for Accurate and Fast Saliency Detection (ITSD) Zhou et al. [2020b] explores the correlation between saliency (global) and contour (local) information by introducing interactive connections between the saliency and contour streams in the decoder. An adaptive contour loss is also introduced in the saliency stream to weight more on near boundary pixels of the salient objects, thus improves the network capability to handling the salient object boundary. The method was evaluated on six publicly available saliency datasets and encouraging results were reported.

# 2 Scope of reproducibility

The investigated work is considered lightweight and able to achieve real-time inference speed at about 88 FPS based on our evaluation. Furthermore, the reported results on six benchmark datasets are also quite encouraging, which achive comparable performance as compared to several SOTA methods. Our reproduction of work consists of the following:

- Reproduce the reported results using the pretrained models as made available by the authors

- Reproduce the reported results by training the models following the exact protocol described in the original paper

- Evaluation of various parameters as reported in the original paper

# 3 Methodology

The authors have made the source code associated with the paper publicly available on GitHub as well as links to download their pretrained VGG-based and ResNet-based models Zhou et al. [2020a]. In order to properly load the model weights, a minor modification is expected to make to the name of the parameter keys in the given PyTorch models as the parameter names are inconsistent with the model definition. To retrain the ITSD models, we followed the training protocol as described in the original paper. The VGG and ResNet, which pretrained on ImageNet dataset, were employed as the backbone of the model. Data augmentation such as random flipping, random cropping and scaling was applied on the DUTS-TR dataset with the aim to increase variability in the dataset. The models were trained on two GTX2080TI GPUs using SGD at the learning rate of 0.01 for the first 20k iterations and decayed by a factor of 0.1 for the remaining 5k iterations.

## 3.1 Model descriptions

The ITSD employs an encoder-decoder network architecture. The backbone of the encoder can be either of a pretrained VGG or ResNet. In order to keep the model lightweight, two measures are adopted: (1) the fully-connected (FC) layers in the standard VGG network is truncated with the aim to reduce the model size while there is no such need for the ResNet-based encoder since it has no FC layer; (2) channel pooling is applied to each of the encoded feature maps. To better recover the saliency map, the decoder of the network employs a two-stream structure, which comprises of five cascaded Feature Correlation Module (FCF) modules. The FCF provides interactive connection (fusion stream) which allows the respective saliency and contour information to be transmitted across the stream, and therefore encourages the network to learn the correlation. To generate the final saliency map, all of the intermediate features from the saliency stream are first up-sampled to the input image size followed by concatenation operation. An adaptive contour loss (ACT) which leverages on the contour information is proposed to provide adaptive weight to the saliency stream, which allows the model to pay more attention to those pixels near to the salient object boundary.

## 3.2 Datasets

Following the original paper, six commonly used publicly available benchmark datasets, namely, DUTS, SOD, PASCAL-S, ECSSD, HKU-IS and DUTS-OMRON were employed in our reproduction work. The number of images for each dataset is summarized in Table 1.

| Datasets | Number of Images | Remarks |
|---|---|---|
| DUTS-TR | 10553 | Training |
| DUTS-TE | 5019 | Testing |
| SOD | 300 | |
| PASCAL-S | 850 | |
| ECSSD | 1000 | |
| HKU-IS | 4447 | |
| DUTS-OMRON | 5168 | |

Table 1: Training and testing datasets

## 3.3 Hyperparameters

As our primary object was to evaluate the reproducibility of the investigated work, we employed the default parameters reported in the original paper, *i.e.*, input image size $= 288 \times 288$, $\lambda = 1$, $m = 4$, and batch size $= 8$, to train the ITSD models. In additional, we also carried out further analysis on various parameters as reported in the paper.

## 3.4 Experimental setup

All the experiments conducted in this work followed the exact protocol as described in the original paper, which we hoped to mitigate the possible differences in some software libraries/packages, in order to reproduce the results as similarly to the reported values. We used the recommended settings from the original paper, as also described in Secs. 3 & 3.3 for the training. Furthermore, we used the source code as made publicly available by the authors to perform both the training and the evaluation, which ensure there exists no bias in the implementation. All the training and evaluation were performed on the same machine with two Nvdia GTX2080TI GPUs, except for the inference speed evaluation which was done using single GPU. Our source code is also made publicly available at GitHub.

## 3.5 Computational requirements

The ResNet-based and VGG-based ITSD models had different requirements on the computational expectations. According to our evaluations, the times required to perform the training and validation were around 5 and 7.5 hours, respectively for ResNet-based and VGG-based models. The overall required time to perform training and validation was observed to be shorter for the ResNet-based model, which may be attributed to its efficient bottleneck architecture.

# 4 Results

In this section, we provide our evaluation results obtained from: (1) the pretrained models as shared by the authors; (2) the retrained ITSD models according to the described protocol and (3) evaluation of various parameters. We presented the evaluation results using two commonly used performance metrics, *i.e.,* $F_{\beta}$ and $MAE$.

## 4.1 Performance evaluation using pretrained models

As can be observed from Table 2 and 3, the evaluation results in terms of $F_{\beta}$ were close to the reported values, with the neglectable differences of up to 1%. However, the degree of differences in terms of $MAE$ was quite noticeable, with the difference as high as 11.4% was observed.

## 4.2 Performance evaluation using retrained models

Table 4 and 5 summarize our evaluation results using the retrained ITSD models. It can be observed that the degree of differences can be as high as 2.4% in $F_{\beta}$ and 8.6% in MAE. In order to quantitatively justifying the reproducibility of

| Datasets | $F_\beta$ | | | MAE | | |
|---|---|---|---|---|---|---|
| | Our results | Reported | Difference (%) | Our results | Reported | Difference (%) |
| ECSSD | 0.946 | 0.947 | 0.11 | 0.034 | 0.035 | 2.86 |
| DUTS-TE | 0.885 | 0.883 | 0.23 | 0.039 | 0.041 | 4.88 |
| DUT-OMRON | 0.819 | 0.824 | 0.61 | 0.059 | 0.061 | 3.28 |
| PASCAL-S | 0.876 | 0.871 | 0.57 | 0.064 | 0.071 | 9.86 |
| SOD | 0.872 | 0.880 | 0.91 | 0.092 | 0.095 | 3.16 |
| HKU-IS | 0.936 | 0.934 | 0.21 | 0.029 | 0.031 | 6.45 |

Table 2: Evaluation results using the pretrained ResNet-based model

| Datasets | $F_\beta$ | | | MAE | | |
|---|---|---|---|---|---|---|
| | Our results | Reported | Difference (%) | Our results | Reported | Difference (%) |
| ECSSD | 0.939 | 0.939 | 0.00 | 0.039 | 0.040 | 2.50 |
| DUTS-TE | 0.880 | 0.877 | 0.34 | 0.041 | 0.042 | 2.38 |
| DUT-OMRON | 0.805 | 0.813 | 0.98 | 0.059 | 0.063 | 6.35 |
| PASCAL-S | 0.871 | 0.871 | 0.00 | 0.067 | 0.074 | 9.46 |
| SOD | 0.860 | 0.869 | 1.04 | 0.106 | 0.100 | 6.00 |
| HKU-IS | 0.931 | 0.927 | 0.43 | 0.031 | 0.035 | 11.43 |

Table 3: Evaluation results using the pretrained VGG-based model

105 the results, we carried out statistical analysis using paired $t$-test at significance level of 0.05 to verify our null hypothesis
106 if there was no significant difference between our evaluation results and the reported results in the article. The paired
107 t-test statistics value can be calculated as follows:

$$t = \frac{\bar{d}}{s/\sqrt{n_d}} \qquad (1)$$

108 where $\bar{d}$ is the mean differences, $s$ is the standard deviation and $n_d$ is the number of observations, *i.e.*, number of the
109 employed benchmark datasets. The $p$-value obtained from the calculated $t$-score is given in the Table 6.

| Datasets | $F_\beta$ | | | MAE | | |
|---|---|---|---|---|---|---|
| | Our results (best epoch result) | Reported | Difference (%) | Our results (best epoch result) | Reported | Difference (%) |
| ECSSD | 0.945 (0.947) | 0.947 | 0.21 | 0.035 (0.034) | 0.035 | 0.00 |
| DUTS-TE | 0.883 (0.883) | 0.883 | 0.00 | 0.040 (0.039) | 0.041 | 2.44 |
| DUT-OMRON | 0.823 (0.824) | 0.824 | 0.12 | 0.057 (0.055) | 0.061 | 6.56 |
| PASCAL-S | 0.870 (0.872) | 0.871 | 0.11 | 0.068 (0.066) | 0.071 | 4.23 |
| SOD | 0.859 (0.875) | 0.880 | 2.39 | 0.099 (0.091) | 0.095 | 4.21 |
| HKU-IS | 0.935 (0.935) | 0.934 | 0.11 | 0.029 (0.029) | 0.031 | 6.45 |

Table 4: Evaluation results using the retrained ResNet-based model

| Datasets | $F_\beta$ | | | MAE | | |
|---|---|---|---|---|---|---|
| | Our results (best epoch result) | Reported | Difference (%) | Our results (best epoch result) | Reported | Difference (%) |
| ECSSD | 0.937 (0.941) | 0.939 | 0.21 | 0.039 (0.037) | 0.040 | 2.50 |
| DUTS-TE | 0.874 (0.879) | 0.877 | 0.34 | 0.042 (0.041) | 0.042 | 0.00 |
| DUT-OMRON | 0.801 (0.804) | 0.813 | 1.48 | 0.062 (0.059) | 0.063 | 1.59 |
| PASCAL-S | 0.869 (0.875) | 0.871 | 0.23 | 0.068 (0.066) | 0.074 | 8.11 |
| SOD | 0.863 (0.866) | 0.869 | 0.69 | 0.103 (0.093) | 0.100 | 3.00 |
| HKU-IS | 0.929 (0.932) | 0.927 | 0.22 | 0.032 (0.030) | 0.035 | 8.57 |

Table 5: Evaluation results using the retrained VGG-based model

| | $p$-value | |
|---|---|---|
| | ResNet-based | VGG-based |
| $F_\beta$ | 0.2956 | 0.1050 |
| MAE | 0.4261 | 0.3276 |

Table 6: Statistical test at significance level of 0.05

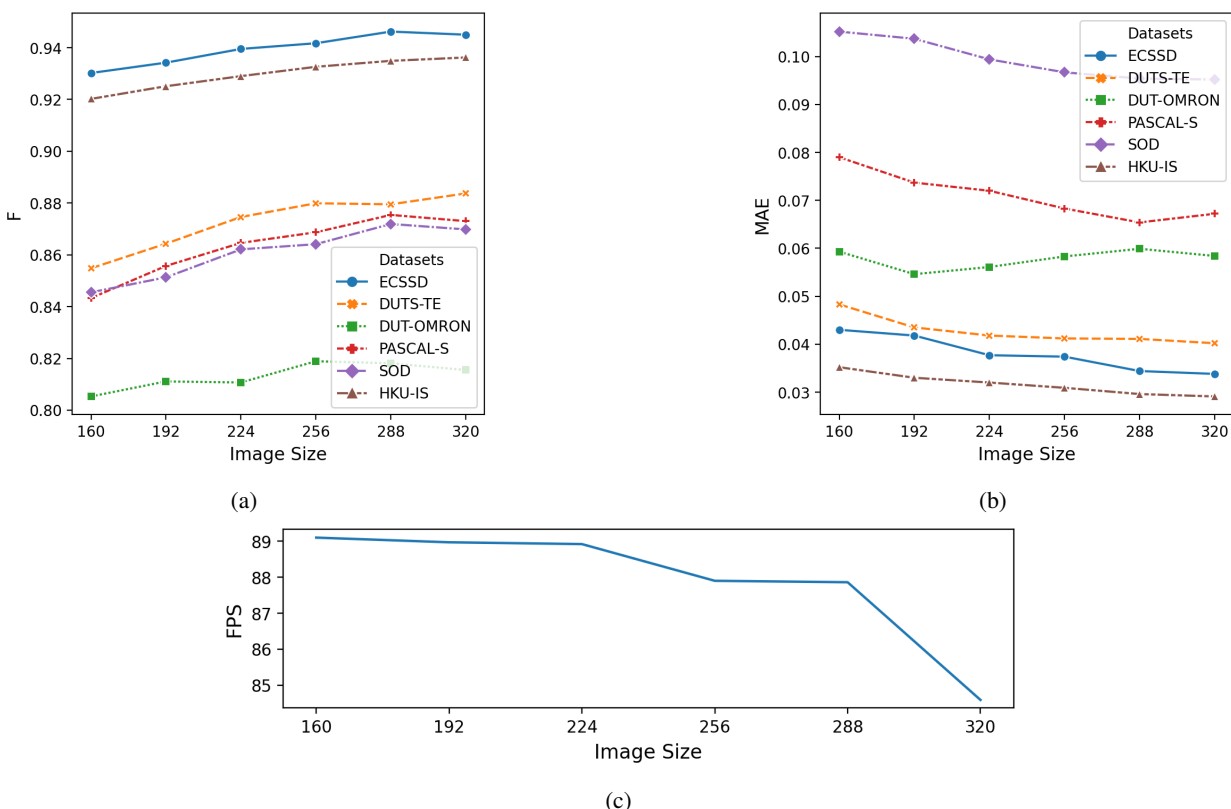

Figure 1: Performance of different input image sizes and inference speed. The recommended image size $288 \times 288$ appear to be a better choice which balances the saliency and the inference performance. (a) $F_\beta$. (b) MAE (c) inference speed.

### 4.3 Additional results not present in the original paper

In this section, we provide further analysis on various parameters that had not been elaborated in the original paper to provide more insights which may be deemed benefit for those working in the field. All the experiments conducted in this section employed the ResNet-based ITSD model, with all the training parameters kept unchanged (see Section 3.3) unless specified otherwise.

#### 4.3.1 Image sizes and inference speed

We have seen different input image sizes were employed for the saliency detection, such as Liu et al. [2019], Zhao et al. [2019], Qin et al. [2019]. Ref. Zhou et al. [2020b] arbitrary chosen an input image size of $288 \times 288$ which motivates us to determine the impact of varying input image sizes on the performance of the ITSD model. We performed the evaluation using the models which trained with input image sizes of $160 \times 160, 192 \times 192, 224 \times 224, 256 \times 256, 288 \times 288, 320 \times 320$. Generally, the performance was better by using larger input image size, as can be observed from the evaluation results as shown in Fig. 1(a) and (b). In addition, we also provide the evaluation of inference speed with different input image sizes on a single GPU, as can be observed from Fig. 1(c). The default input image size $288 \times 288$ reported the inference speed ran at around 88 FPS while the inference speed for the input image size of $160 \times 160$ achieved 89 FPS, at the expense of the saliency performance. The input image size of $288 \times 288$ emerges to be a better choice which balances the saliency performance and the inference speed.

#### 4.3.2 Hard examples factor $m$

The hard examples factor $m$ in the saliency loss function serves to emphasize the boundary pixels so the model can better handling the boundary of the salient regions. However, the effect of such parameter had not been discussed in the

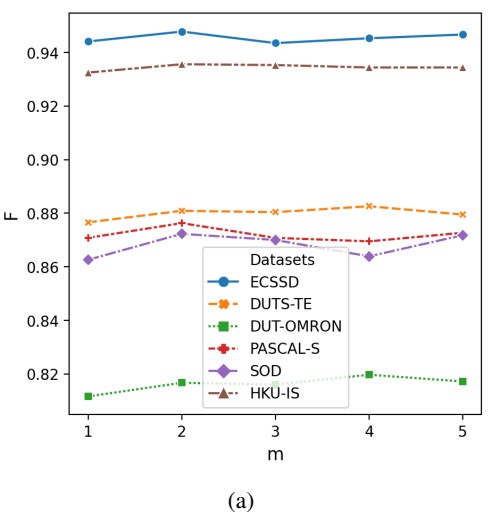
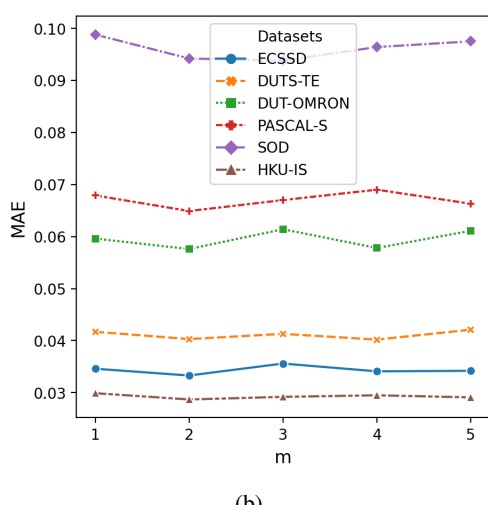

|  | (a) | (b) |

Figure 2: Effect of different hard example factor (a) $F_\beta$. (b) MAE

original paper. Fig. 2 shows our evaluation results of using various models respectively trained with $m = \{1, 2, 3, 4, 5\}$. On average, some marginal gains were observed in both metrics with the hard examples factor $m = 2$.

### 4.3.3 Dilation and erosion

The contour map is obtained by computing the difference between the saliency maps produced by two morphological operations, *i.e.*, dilation and erosion. The resultant from such operation is a contour map with the border width determined by the morphological kernel size, which in turn affecting how the model learn about the hard example pixels near the salient object boundary. In the original authors' implementation, morphological kernel size of $5 \times 5$ was used. In order to study the effect of different morphological kernel sizes, we considered $1 \times 1, 3 \times 3, 5 \times 5, 7 \times 7, 9 \times 9$ in our experiments. Fig 3 shows samples of contour maps generated using different morphological kernel sizes. Note that no contour information is produced by using the morphological kernel size of $1 \times 1$ and therefore the Eq. 12 (refer to original paper) will be solely depend on the $P^c$ to weight the saliency loss function. As can be observed from Fig. 4, the morphological kernel size of $3 \times 3$ can generally provide better performance than the default kernel size, but such improvement are only marginal.

### 4.3.4 Lambda, $\lambda$

The $\lambda$ in the total loss function serves to balance between the saliency and contour losses, which was set to $\lambda = 1$ in the original paper. In other words, one can determine how much emphasis to be placed on the contour loss by adjusting the $\lambda$. In our experiments, we performed evaluation for a range of values, *i.e.*, $\lambda = [0.0, 1.6]$, with a step size of 0.2. Based on the average results from our evaluation, the $\lambda = 0.6$ had shown to achieve better performance, which suggested that the model is biased towards the saliency information as such information is more crucial, while complemented by the contour information for further performance gain. The $\lambda = 0$ totally suppresses the contour loss, which can be considered as a special case of the total loss function, where only saliency information is considered.

### 4.3.5 Optimized vs. default parameters

|  | Optimized | Default |
|---|---|---|
| Input image size | $288 \times 288$ | $288 \times 288$ |
| Morphological kernel size | $3 \times 3$ | $5 \times 5$ |
| Hard examples factor, $m$ | 2 | 4 |
| Lambda, $\lambda$ | 0.6 | 1 |

Table 7: Optimized vs. default parameters

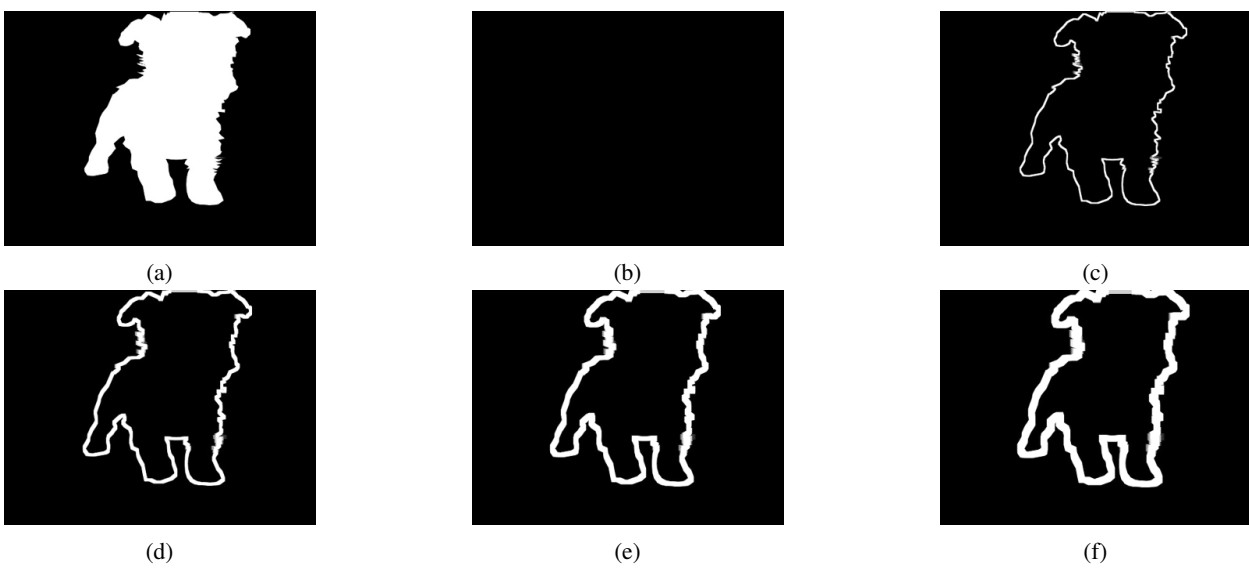

Figure 3: Contour maps generated using different morphological kernels (a) mask, (b) $1 \times 1$, (c) $3 \times 3$, (d) $5 \times 5$, (e) $7 \times 7$, (f) $9 \times 9$

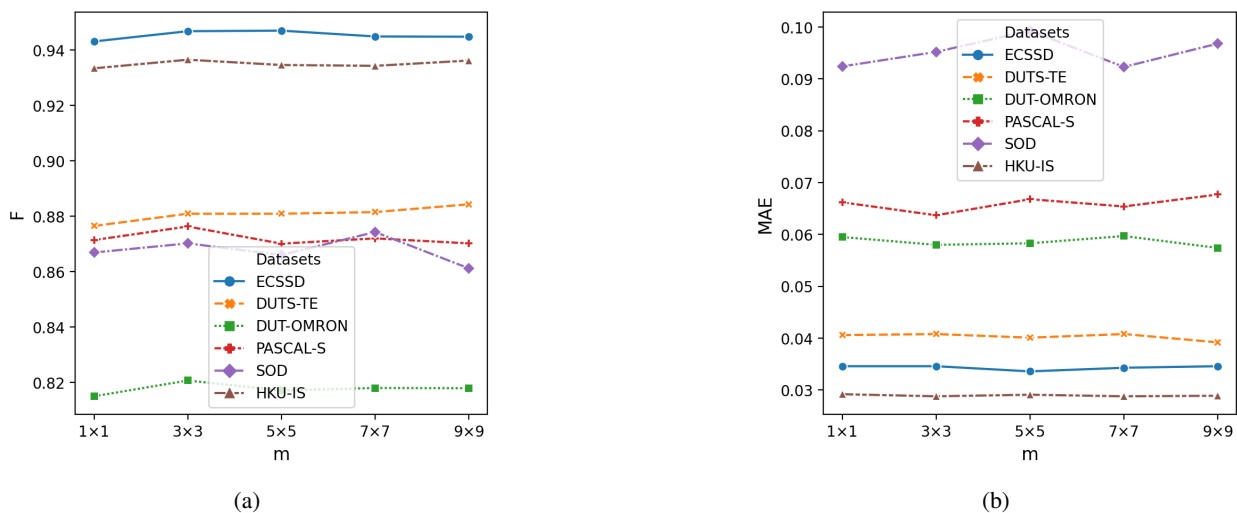

Figure 4: Effect of morphological kernel size (a) $F_\beta$. (b) MAE

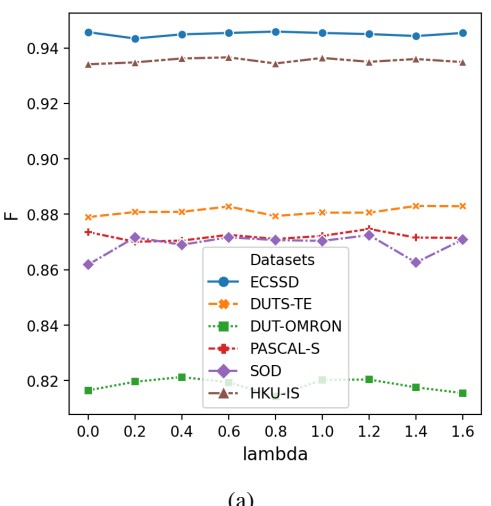
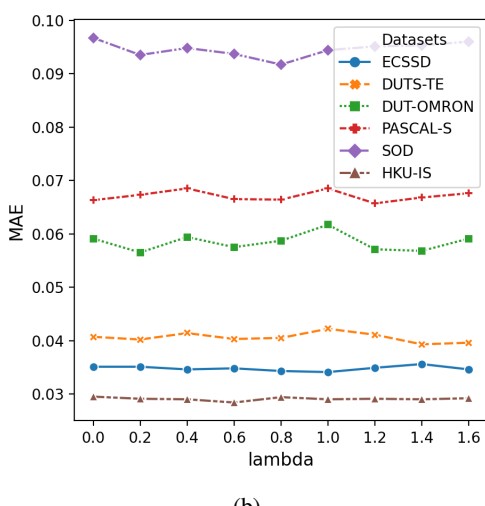

|||
|:-:|:-:|
|(a)|(b)|

Figure 5: Effect of lambda (a) $F_\beta$. (b) MAE

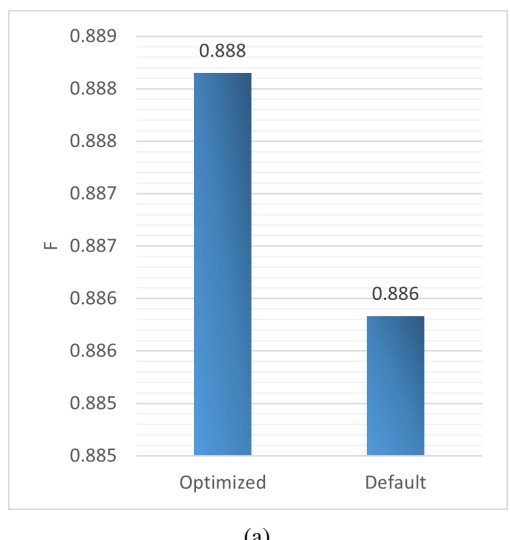
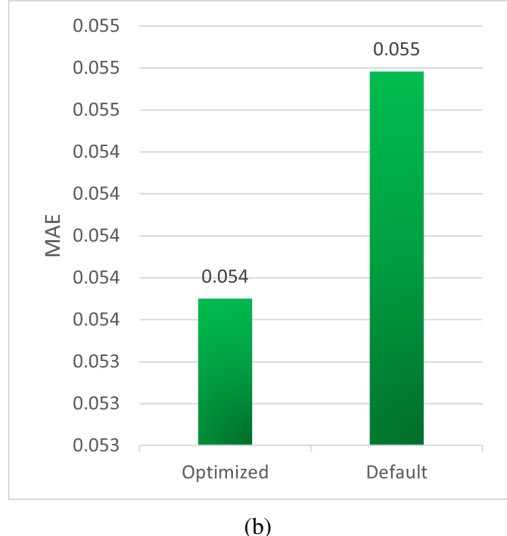

|||
|:-:|:-:|
|(a)|(b)|

Figure 6: Marginal performance improvement is observed with our optimized parameters as compared to the default parameters. (a) $F_\beta$. (b) MAE

Sections 4.3.1 - 4.3.4 show our evaluations on the various parameters of the ITSD model, and each individual parameter has been shown to induce certain degree of improvement on the average performance metrics. As such, we carried out experiment by using the optimized parameters, as summarized in Table 7. Fig. 6 presents the average results of the comparison between the models trained with optimized and the default parameters, which only marginal gain was observed. Our primary intention to carry out the experiments in Sections 4.3.1 - 4.3.4 was to study the implication of each parameter towards the saliency detection performance. Therefore, each of the experiment conducted only evaluated the parameter of concerned while other parameters were kept unchanged in order to better facilitate the analysis. One should consider a proper parameter optimization approach such as grid search to exhaustively searches through the parameter spaces to better estimate the best parameters' combination.

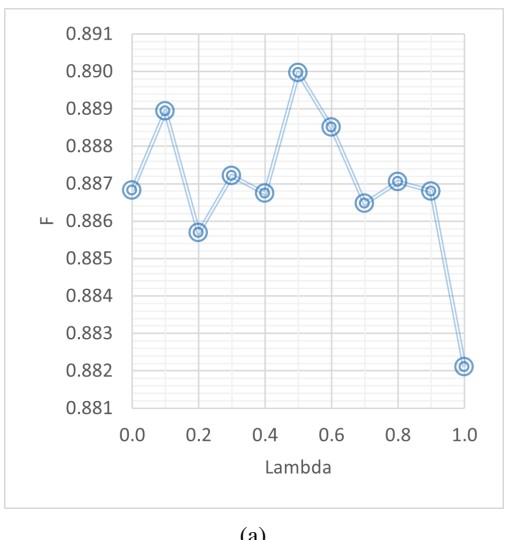
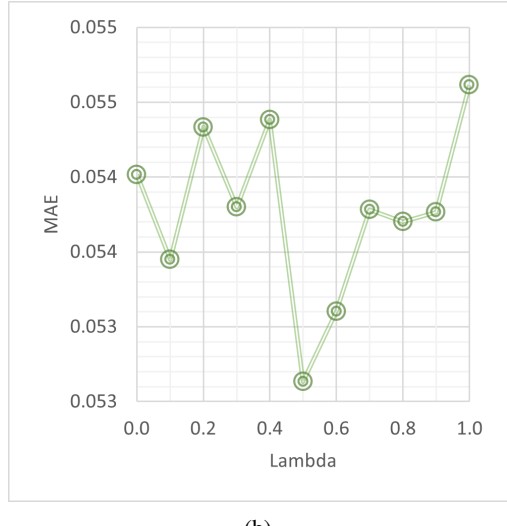

Figure 7: Constrained lambda. (a) $F_\beta$. (b) MAE

 ## 4.4 Constrained lambda

Although the free parameter, *i.e.,* $\lambda$ in the total loss function was analyzed in Section 4.3.4, it is still uncertain which of the losses contribute more towards the optimization of the network. In order to further analyze the importance of each loss term, we imposed constraint to the toal loss function by considering a weighted sum rule, as given as follows:

$$L(P^s, P^c, G^s, G^c) = \sum_{i=0}^{5}(1 - \lambda)L^s(P_i^s, P_i^c, G_i^s, G_i^c) + \lambda \sum_{j=1}^{5} L^c(P_j^c, G_j^c) \tag{2}$$

s.t. $0 \le \lambda \le 1$

By constraining the lambda would enable us to analyze the correlation between the saliency and contour losses, as can be observed from average performance metrics in Fig. 7. The best saliency detection performance was achieved when the $\lambda = 0.5$, which suggested that both the saliency and contour losses contributed equally and complemented each other. While the worst performance was reported when the $\lambda = 1.0$, which implied that the contour loss alone appeared to be suboptimal for optimizing the model.

## 5 Discussion and conclusion

We had some issues to get the published code working initially but those issues were solved after some discussion with the authors. One of the major issues was the inconsistent use of the name of the parameter keys in the given PyTorch models, which resulted in failing to load the model weight. The issue can be solved by updating the name of the parameter keys to be compatible with the defined models.

Despite of all the challenges we faced during the work reproduction, we eventually managed to reproduce the work of the investigated paper, *i.e.,* ITSD. Our reproduction attempts were mainly comprised of two parts: (1) reproduction using the pretrained models as shared by the authors; (2) reproduction by training the models. Our attempt using the pretrained models was of no success to reproduce the exact results as reported in the original paper, as can be observed in Table 2 and 3. After seeking clarification from the authors, one of the possible causes of such discrepancy may be due to the environment settings. Another possible reason may be due to the best epoch results were reported in the paper, while the shared models represented the final training artifacts. Our assumption was evidenced by carrying out the experiments to capture the best epoch results, which we successfully obtained either similar or better performance as compared to the reported results. Our second reproduction attempt was to perform the retraining of the ITSD models. Despite differences in the experimental results, we carried out statistical test, *i.e.,* paired $t$-test, to further confirmed the

reproducibility of the presented work in the studied paper. Further to our reproduction attempts, we had also carried out experiments to perform evaluation on the various parameters which had not been elaborated in the original paper. We hope that our reproduction work can be beneficial to those working in the same direction, as well as serving as a recognition to the contributions of the original authors.

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
