# OpenReview forum: "[Re] Interactive Two-Stream Decoder for Accurate and Fast Saliency Detection"
_ML_Reproducibility_Challenge/2020 — Reject_

### Official Review · AnonReviewer3 · 2021-03-01
**Interactive Two-Stream Decoder for Accurate and Fast Saliency Detection**

**Rating:** 7
**Confidence:** 3

**Review:**

***Reproducibility Summary:***
       The authors have provided a detailed summary meeting the requirements of a reproducibility report.

***Scope of reproducibility:***
        Yes, the reproducibility report has clearly and concisely stated the scope of reproducibility.

***Code:***
        Yes, the authors have re-used the original author's code repository and their trained model with small modifications.

***Communication with original authors***
      Yes, the authors connected with original authors through the original authors' Github repo.

***Hyperparameter Search:***
      Yes, the authors have attempted to reproduced the hyperparameter search.  The authors have also expanded the hyperparameter search to involve image size, the "hard examples factor $m$", different kernel sizes, lambda parameters, and other analysis not present in the original paper.

***Ablation Study:***
      I did not notice any ablation in the study.

***Discussion on results:***
      Yes, the reproducibility report contains a brief discussion on the state of reproducibility of the original papers, but also provides  notes on where to modify the original work to fix the issue in the original pyTorch model.  Despite differences in the experimental results, the authors carried out a paired t-test to further confirm the reproducibility of result.

***Recommendations for reproducibility:***
      No, the authors did not provide any recommendation to the original authors for improving reproducibility.

***Results beyond the paper:***
      The authors have tried additional results that are not mentioned in the original paper.  The authors include significantly more quantitative and qualitative results than the original paper.

***Overall organization and clarity:***
         Nicely written and coherent.

***Pros:***
         Significantly more quantitative and qualitative results.
         Extensive further exploration of the hyperparameter search and added new dimensions to the hyperparameter search (e.g. image size).

**Familiar With The Original Paper:**

I have not read the original paper

**Reproducibility Summary:**

Report has summary

---

### Official Review · AnonReviewer1 · 2021-03-01
**Review on Paper 7**

**Rating:** 6
**Confidence:** 4

**Review:**

Authors of this report provide summary of report, scope of reproducibility and communicated with original author of the “Interactive Two-Stream Decoder 3 for Accurate and Fast Saliency Detection”.

This paper consists of mainly three parts: evaluation using the pretrained model, evaluation using retrained model and hyperparameter ablations

Basically, there are available codes from the original authors, thus it is important to perform plenty of “hyperparameter search” and “ablation”.

The good point of the paper is that the author tried to verify the original model provided by the original authors and reproduce and retrain the exact same model, then perform ablations. It is helpful for the following works.


In order to quantitatively justify the reproducibility of the results from the retrained model, they carried out statistical analysis using paired t-test. I think that it is a good value of this report.

Also, this report provides more various ablation studies not included in the original paper, including “image size and inference speed”, “factor m”, “dilation and erosion”, “lambda”, etc. Those plenties of experiments may be helpful for following researchers.


**Familiar With The Original Paper:**

I have read the original paper

**Reproducibility Summary:**

Report has summary

---

### Official Review · AnonReviewer2 · 2021-03-02
**Reasonable RC report**

**Rating:** 6
**Confidence:** 3

**Review:**

The scope definition is good and clear.

The design choice in choosing training and testing datasets would need a bit better motivation.

This paper did make efforts in exploring different hyperparameter searches.

There are communications between authors and the original authors. All are done in Github issues. I am happy to see that the authors have made a few changes to the original code.

This paper picked up some experiments that are not done in the original papers. e.g. effects of image sizes and inference speed; hard examples factor m; dilation and erosion; lambda; optimized vs default parameters; constrained lambda.

It would be better if the authors could look further into the compatibility of the pre-trained models with other frameworks e.g. TensorFlow, or ONNX.

I find the following items could be improved in the paper.
1. Table 2-6 lacks the detailed description of "our results". How exactly the parameters are set to get the results?
2. I think figure 1 could be significantly improved if there are more descriptions to summarize the trends in the figures.
3. Figure 3 is a bit confusing, how the figures related to the proposed approach?
4. Figure 6 provides very little information, and it should be replaced by a table or even description in the text.

**Familiar With The Original Paper:**

I have read the original paper

**Reproducibility Summary:**

Report has summary

---

### Decision · Program_Chairs · 2021-03-31

**Decision:**

Reject

**Comment:**

Overall reviews and/or the paper content not good enough for the AC to recommend to the journal.